# Properties of Vermicomposts Derived from Cameroon Sheep Dung

**Mariola Garczyńska** [1,*] , **Joanna Kostecka** [1], **Grzegorz Pączka** [1], **Edmund Hajduk** [2] ,
**Anna Mazur-Pączka** [1] **and Kevin Richard Butt** [3]

1   Department of the Basis of Agriculture and Waste Management, Institute of Agricultural Sciences, Land
    Management and Environmental Protection, College of Natural Sciences, University of Rzeszow,
    Cwiklinskiej 2, 35-601 Rzeszow, Poland; jkosteck@ur.edu.pl (J.K.); grzegp@ur.edu.pl (G.P.);
    anamazur@ur.edu.pl (A.M.-P.)
2   Department of Soil Science, Chemistry of Environment and Hydrology, Institute of Agricultural Sciences,
    Land Management and Environmental Protection, College of Natural Sciences, University of Rzeszow,
    Cwiklinskiej 2, 35-601 Rzeszow, Poland; ehajduk@ur.edu.pl
3   School of Natural Sciences, University of Central Lancashire, Preston PR 1 2HE, UK; krbutt@uclan.ac.uk
*   Correspondence: mgar@ur.edu.pl; Tel.: +48-17-872-1649

**Abstract:** Due to a need for sustainability in agriculture, waste products ought to be utilized in the
most appropriate way. A study was undertaken relating to the vermicomposting of Cameroon sheep
dung (CSD) by the earthworm *Dendrobaena veneta*. Processing of this waste was investigated using
unadulterated CSD and in a 1:1 mixture with unpalatable (waste) hay (CSDH). Results demonstrated
that these materials were actively processed by *D. veneta* with vermicomposts obtained which can be
characterized by a higher amount of total nitrogen, phosphorus and potassium (average, respectively,
17.0, 10.5, 13.2 g kg$^{-1}$ d.m.), as well as lower total carbon and magnesium content (respectively,
340 and 3.2 g kg$^{-1}$ d.m.), compared with the initial waste material. No significant differences were
found between CSD and CSDH vermicomposts with respect to chemistry. Levels of selected trace
elements (average: Cu 17.5–18.8, Cr 5.7–5.8, Pb 13.5–14.4, Ni < 3, Cd < 0.4 mg kg$^{-1}$ d.m.) in both
vermicomposts did not exclude their application to agricultural soil as a fertilizer.

**Keywords:** Cameroon sheep dung; earthworms; *Dendrobaena veneta*; hay; macroelements; trace
elements; vermicomposting

## 1. Introduction

A sustainable agriculture strategy demands the implementation of multidirectional actions that
combine productive aims of agriculture with environmental protection requirements. Soil protection is
one of the significant objectives within this area. European Union countries are obliged to maintain
appropriate soil functioning and take essential steps to avoid major processes that degrade the soil
environment, stop their intensification and determine regions where degradation occurs [1]. A decrease
in organic matter content and depletion of nutrients are the major threats to the natural functioning
of soils. Loss of organic matter causes disturbances of chemical, physical and biological processes
and alters the biogeochemical cycle of many elements which leads to the deterioration of quality and
productivity of soils [2–5]. Analyzing the problems of soil ecosystems, The International Decade of Soils
(2015–2024) was announced by The International Union of Soil Sciences (IUSS) at a Soil Conference in
Vienna (5 December, 2015) [6].

One of the many factors causing such a situation is the use of mineral fertilizers. In Poland, an
average of 133 kg ha$^{-1}$ is used on agricultural land [7]. In 2017, 11.6 million tons of nitrogen fertilizer

was used in EU agriculture—an increase of 8% since 2007 [8]. The use of mineral fertilization is inherent in additional costs, especially the purchase of nitrogen and phosphorus fertilizers is relatively expensive. In addition, some of them (mainly P or Ca) can be a source of soil pollution with heavy metals [9,10]. Analysis of soil organic matter content and other fertility parameters suggest it would be more valuable to apply organic fertilizer or a mixture of organic and mineral fertilization [11,12].

An increase in *Homo sapiens* population, urbanization, industrialization and agriculture has increased the production of wastes, including those that are biodegradable. Of the latter, animal dung, may constitute a high-quality resource, serving for the production of composts and vermicomposts [13,14]. Among these resources are manures from horses, cattle, pigs, poultry and other animals [15–20]. Manure used as a fertilizer can be a source of many nutrients for plants [21–24]. For example, the analysis of the chemical composition of various types of manure in Poland showed on average 1 kg of dry matter: 4.8 g N, 3.1 g $P_2O_5$, 6.7 g $K_2O$, 4.3 g CaO, 1.6 g MgO, 21.5 mg B, 23 mg Cu, 194 mg Zn, 1.63 mg Mo, 345 mg Mn, 2449 mg Fe [24]. The application of manure-derived vermicomposts has an effect on soil fertility and crop quality [18,25–27]. This is particularly important in light soils, in which the water–air relationships cause naturally lower organic matter content compared to heavy soils [5].

In recent years, an interest in alternative stock of farmyard species has systematically grown in Poland [28]. This includes hair sheep, for example, Cameroon sheep or Somali sheep. Hair sheep originate from Africa but have acclimatized well to Polish conditions. Properly managed excrement, or vermicompost produced from hair sheep, could be an additional benefit for a sheep farmer. This farm-produced organic fertilizer can be used for plant growth and improvement of soil quality. Although there is literature on vermicompost from general sheep dung [29–31], no characteristics have been produced for the vermicomposting of Cameroon sheep droppings.

In addition, it is worth noting as an issue, that under South Eastern Poland's conditions, agri-environmental programs are very popular [32]. Farmers covered by these programs, postpone mowing of meadows until mid-July or even after 1st August, to allow ground-nesting birds to successfully raise young. Such action, however, reduces the fodder value of the hay collected thereafter, which is only reluctantly eaten by stock. Such hay can be used as bedding, stored for natural biodegradation or burnt in biomass boilers. Nevertheless, one limitation in the production of pellets or biomass from hay is the physico-chemical composition as it contains numerous difficult-to-degrade substances (suberin or cutin). A viable alternative could be utilization in a vermicomposting process, following recommendations of the General Directorate for Environmental Protection (2015) [33] and the Ministry of Agriculture and Rural Development (2019) [34].

Interest in sheep dung is a result of subsidized sheep breeding, supported by the Rural Development Programme for the period 2015–2020, which opens up the characterization of this manure as another reason for the study. Fertilizers are defined by Polish law in the Fertilizers and Fertilization Act of 7 June, 2018 [35] as products intended for providing plants with nutrients or increasing soil fertility, which are mineral fertilizers, natural fertilizers, organic fertilizers and organic-mineral fertilizers. This Act [35] indicates the process of vermicomposting as an appropriate method for the production of valuable organic fertilizers (chapter 1, article 2.1 item 5). Characteristics that support the use of vermicompost are high water capacity, high porosity and gradual, slow release of nutrients to the soil [36–38]. The content of macroelements in composted manure is usually higher than fresh animal manure. Composting of animal manure also encourages soil microbial activity, which promotes the soil's trace mineral supply, improving plant nutrition [39]. If compared with unprocessed manure, it helps with the stabilization of the soil profile, prevents erosion, improves soil structure, is favorable to retaining moisture and can reduce problems with drainage in damp areas. Additionally, manure may contain pathogenic organisms and constitute a potential source of zoonoses, and therefore it may be the main risk for spreading diseases among animals and people [40]. Numerous studies highlight the effectiveness of vermicomposting manure of different origins to reduce pathogenic microorganisms. A reduction in pathogens occurs by passage through the intestines of an earthworm, which obviates

the need to raise the temperature. The final product, with regard to microbiological quality, can then be considered as class A compost [41–44].

In addition, vermicomposts have been shown to provide a favorable influence on growth, development, flowering, fructification and plant health. They also limit the expansion of fungal diseases, such as *Pythium*, *Rhizoctonia*, *Plectosporium* and *Verticillium;* the presence of numerous pests, nematodes in field cultivation of tomatoes, peppers, strawberries and grapes, as well as *Myzus persicae* aphid, which is a transmitter of many viruses; and *Pseudococcus* and *Pieris brassicae* caterpillars in the cultivations of tomatoes, peppers and cabbages. Such positive impacts of vermicomposts, in field experiments and greenhouse cultivation, has been shown by many authors [30,45–56]. By applying vermicompost, soil can be enriched with organic matter and mineral compounds, to be absorbed by plants. Besides this, vermicompost fertilizer also causes an increase in biological activity in the soil and improves structure [30]. Determination of the main macro-elements (nitrogen, carbon, potassium, phosphorus, calcium, magnesium) of the soil is one of the most important criteria of the assessment of soil richness [57].

The major aim of the current research was to determine the feasibility to vermicompost Cameroon sheep dung, in a pure form and mixed with marginal value hay from agri-environmental programs. A further aim was to determine the properties of vermicomposts obtained from these droppings. Finally, the effects of these substrates on growth of the populations of the vermicomposting species *Dendrobaena veneta* were also to be examined.

## 2. Materials and Methods

### 2.1. The Source of the Vermicomposted Substrate—Cameroon Sheep Dung

Cameroon sheep dung was collected from an agritourism farm in Krosno. It consisted of waste from these animals and cereal straw litter. The feces were dried at a temperature of 105 °C ready for use. This was to reduce the ammonium levels known to be detrimental to some earthworms.

### 2.2. Vermicomposting

The experiment was conducted at the University of Rzeszów in 2016–2017 and ran for a period of six months in each year.

Earthworms used in the experiment (*Dendrobaena veneta* Rosa, 1886) came from our own multi-annual breeding line. Before starting the experiment, the earthworms were selected from breeding beds and placed into containers filled with garden soil and manure for 14 days, so that they would become acclimatized to experimental conditions, in accordance with the methodology used by Kostecka et al. [15].

The experiment was carried out in containers: $18 \times 18 \times 8$ cm. Each was filled with garden soil (2 dm$^3$) of known physico-chemical characteristics (Kronen universal soil: pH in H$_2$O 6.0–6.5; salinity 1.0–2.0 (mg·dm$^{-3}$); N $355 \pm 57$ (mg·dm$^{-3}$); P$_2$O$_5$ $300 \pm 36$ (mg·dm$^{-3}$); K$_2$O $400 \pm 59$ (mg·dm$^{-3}$) solid, loose form, fraction 0–20 mm) and properly balanced food. Holes in the containers were made to allow for air flow. Waste biomass was contained in traps made of lathing mesh.

Eight containers were divided between two treatments: Cameroon sheep dung (CSD) and CSD mixed in a 1:1 ratio with unpalatable (waste) hay (CSDH) as outlined below:

- CSD (10 mature *D. veneta* with known and balanced biomass were placed within each container—($0.550 \pm 0.016$ g) + garden soil + 400 g of Cameroon sheep dung;
- CSDH (10 mature *D. veneta* with known and balanced biomass within each container—($0.553 \pm 0.011$ g) + garden soil + 400 g of Cameroon sheep dung mixed with dampened hay (in 1:1 ratio).

To maintain appropriate waste humidity, containers were dampened every 7 days with the same volume (100 mL) of tap water (pH—7.5, conductivity—486 µS·cm$^{-1}$, nitrates V—8.5 mg·dm$^{-3}$, nitrates III—0.01 mg·dm$^{-3}$, Mg—14.2 mg·dm$^{-3}$, hardness water—231 CaCO$_3$ mg·dm$^{-3}$·).

The containers were placed in an air-conditioned chamber (temperature $20 \pm 0.5\,°C$, humidity 70%). Macroelement (N, P, K, Ca, Mg) content was determined both in the waste material and in vermicompost, in accordance with the methodology used by the Ministry of Agriculture, Fisheries and Food [58]. All measurements were performed in three replications. Moreover, some trace elements (Ni, Pb, Cr, Cd, Cu) were also determined in the vermicompost.

Organic carbon was determined by use of an elemental analyzer Vario EL Cube (Thermo Fisher Sci); N—by Kjeldahl's method, pH in $H_2O$—by potentiometric method, salinity (NaCl $g \cdot dm^{-3}$) and conductivity (mS)—by conductometric method, phosphorus—by vanadium-molybdenum method, potassium and calcium—by flame photometry method (FES), magnesium—by atomic absorption spectrophotometry method (AAS). The C:N ratio was calculated (in mass/mass ratios).

Cd, Cr, Cu, Pb and Ni were determined in soil extracts with aqua regia—using flame and electrothermal absorption spectrophotometry [59]. In the research pursued, the total content of trace elements was determined. The uncertainty for the total analytical procedure (including sample preparation) was below the level of 15%. The accuracy was checked using reference materials SRM 2709a (San Joaquin Soil) and the recovery (87%–102%) was acceptable for determined heavy metals.

The dynamics of the populations of *D. veneta* during vermicomposting (CSD and CSDH) were assessed 5 times per month, using manual sorting [60]. Earthworms found were counted and had biomasses determined.

The results were developed and presented as the mean $\pm$ SD. Statistical analysis at a confidence level of 0.05 was performed with the use of a Student's t-test using Statistica software v.10.1.

## 3. Results and Discussion

Vermicomposts produced from CSD and CSDH were rich in nutrients (Table 1), but none of these analyzed characteristics were statistically different ($p > 0.05$). Characteristic comparisons of Cameroon sheep dung (initial material) with the vermicomposts obtained, showed differences in salinity, total C, N as well as P, K and Mg ($p < 0.05$).

**Table 1.** Selected features and total macroelement content in base material and in vermicompost.

| Features Units | Characterized Substrates | CSD | CSDH |
|---|---|---|---|
| pH in $H_2O$ | Initial material | 7.4–8.4 | 7.2–7.8 |
| | Final | 6.7–6.8 | 6.7–6.8 |
| conductivity (mS) | Initial material | 2.9 ± 0.3 [a] | 3.1 ± 0.4 [a] |
| | Final | 3.75 ± 0.05 [a] | 3.9 ± 0.1 [a] |
| salinity (NaCl $g \cdot dm^{-3}$) | Initial material | 4.9 ± 0.3 [a] | 4.8 ± 0.3 [a] |
| | Final | 1.2 ± 0.3 [b] | 1.1 ± 0.1 [b] |
| C | Initial material | 480.6 ± 6 [a] | 480.7 ± 5 [a] |
| | Final | 340 ± 10 [b] | 365 ± 14 [b] |
| N | Initial material | 14.0 ± 2 [a] | 14.5 ± 3 [a] |
| | Final | 17 ± 1 [b] | 18 ± 2 [b] |
| P | Initial material | 8.0 ± 0.5 [a] | 8.0 ± 0.4 [a] |
| | Final | 10.5 ± 0.7 [b] | 9.6 ± 0.2 [b] |
| K | Initial material | 10.0 ± 0.6 [a] | 10.4 ± 0.5 [a] |
| | Final | 13.2 ± 0.4 [b] | 13.6 ± 0.4 [b] |
| Ca | Initial material | 15.5 ± 0.2 [a] | 15.7 ± 0.3 [a] |
| | Final | 16.2 ± 0.3 [a] | 16.6 ± 0.3 [a] |
| Mg | Initial material | 5.5 ± 0.5 [a] | 5.5 ± 0.6 [a] |
| | Final | 3.2 ± 0.3 [b] | 3.0 ± 0.3 [b] |
| C/N | Initial material | 34.7 ± 0.4 [a] | 35.1 ± 0.3 [a] |
| | Final | 20.0 ± 0.6 [b] | 20.0 ± 1.0 [b] |

The group spanning rows C, N, P, K, Ca, Mg units: $g\,kg^{-1}$(d.m.)

CSD—final-vermicompost from Cameroon sheep dung (initial material—Cameroon sheep dung); CSDH—final-vermicompost from Cameroon sheep dung with mixed dampened hay in 1:1 ratio (initial material—Cameroon sheep dung with hay). aa, bb—no statistically significant differences, ab—statistically significant differences.

### 3.1. Humus

Organic matter changes in the composting process promote the formation of humus compounds. Humus is formed from the remaining part of detritus during the process of humification. This process has a comprehensive influence on soil, resulting in the improvement of its properties. The addition of composts or vermicomposts (especially to a sandy soil) causes a gradual increase in organic matter content, which in turn improves soil aeration, ensures soil protection against erosion and increases the availability of nutrients [38,61,62]. Humus soil content is different, dependent on the agronomic category of the soil and climatic-soil condition. In Poland, in terms of humus content, mineral soils are divided into low humus soils (below 1%), medium humus soils (1.01%–2%), humus soils (2.01%–3%) and high humus soils (over 3%). It is estimated that up to 2030, on average, 0.8% of organic matter in Poland will be degraded [63]. About 45% of soils in Europe have low or very low organic matter content (0%–2% organic carbon) [8].

The main factors decreasing the amount of humus are intensification of agriculture, unsustainable mineral fertilization and simplified crop rotation. Plants that contribute the most to the depletion of soil organic matter resources are root crops and corn, whereas leguminous plants and their mixtures with grasses enrich the soil the most with organic matter.

### 3.2. Changes in pH

Cameroon sheep dung and vermicomposts produced in our experiment did not differ in pH value ($p > 0.05$). According to Jahanbakhshi and Kheiralipour [30], vermicompost has a more neutral pH compared to sheep dung, and the use of a high amount has no detrimental effects on soil and plants.

Importantly, both manure and the vermicomposts produced had an appropriate pH at which most of the macroelements contained in the soil were available to plants (pH range 6.5–8.5). In addition, the pH of vermicomposts may allow the availability of most micronutrients (e.g., Zn or Cu).

The exposure of ecosystems to acidification in the EU-28 (critical loads from 43% in 1980 to 7% in 2010) and EEA member countries (to 7%) has been decreasing since the 1980s, although in some areas, reduction targets, defined as interim objectives in the EU's National Emission Ceilings Directive, have not been met [64]. The prolonged use of mineral (NPK) fertilization decreases pH value, especially in light soils, therefore organic fertilization may be a better alternative to mineral fertilization [65].

### 3.3. Changes in Salinity

The salinity of vermicomposts, which was significantly lower compared to initial material dung, is a significant factor affecting soil properties as well as the growth and development of plants ($p < 0.05$). Similarly, Aali et al. [29], Jahanbakhshi and Kheiralipour [30] and Alikhani et al. [66] have recorded lower salinity in the vermicompost made in comparison to sheep dung. The reason for this finding might be attributed to the easier leaching in vermicompost and to a lesser degree, by the consumption and accumulation of ions in the earthworm biomass.

If the salinity is too high, it may lead to disturbances in the plant uptake of such macro-elements as P, Mg and N [11,38,67]. Soil salinization is projected to increase in future climate change scenarios due to sea level rise and impact on coastal areas, and the rise in temperature will inevitably lead to increased evaporation and further salinization. Soil salinity is dynamic and spreading globally in over 100 countries [68].

### 3.4. Macronutrient Content

Both the dung and the vermicomposts produced had a high Ca content. This macro-element (Ca) is important for soils and crops. According to the Institute of Soil Science and Plant Cultivation (IUNG) [63], approximately 49.7% of the examined soil samples required liming (including Małopolska Province 77.5%, Podkarpacie Province 73.4% and Kujawy-Pomorze Province 26.5%). In Poland, approximately 60% of arable soils require liming and approximately 20% of soils exhibit Mg, Zn, Mn

and Cu deficiencies [4]. The application of lime depends on the agronomic category of soil (very light, light, medium or heavy).

Deficiency of P in the soil restricts plant growth and development. Vermicomposting increased the content of P for plants in vermicomposts compared to the base material ($p < 0.05$). According to Alidadi et al. [69], during the process of vermicomposting in the gastrointestinal tract of earthworms, the P content increases with the participation of alkaline bacteria. Earthworms change insoluble P to a soluble form via the phosphatase enzyme present in their gut [19]. The increase in available forms of P in vermicompost increase the content of P in the soil at a pH close to neutral. In the case of pH values outside of 5–7, there will most probably be retrograde motion (that is reversible) of easily assimilate forms of P [70].

Coulibaly et al. [18] noticed that vermicompost (generated from manure) caused better assimilation of P by the plant *Lagenaria siceraria* in comparison to crops without fertilization and with cattle manure fertilization. Similar results were obtained by Mahmud et al. [38] in pineapple cultivation where, after six months, he noticed that the P from the vermicompost was the most assimilable.

The content of the total forms of macro-elements, such as K, in vermicomposts significantly increased compared to the base dung ($p < 0.05$). An increased K level in vermicompost indicates the increased rate of mineralization of organic compounds containing K [71]. This is beneficial because light soils, frequently acidified, are very poor in available K. This is also interesting, in the process of vermicomposting, despite the increase in the content of K and a more or less constant level of Ca, the salinity decreased. Salinity was determined on the basis of the specific conductivity of vermicompost suspension in water, so the value of this parameter (and its estimated salinity) is influenced by various ions, including those not tested and not specified in the work. Maybe the primary substrates used for composting were different from K + and Ca 2+ charge carriers—acids or salts, for example in the form of organic compounds—that have been mineralized. This could be an interesting starting point for further studies.

In vermicomposts, compared to the dung, the content of total P increased ($p < 0.05$), whereas Mg content decreased ($p < 0.05$).

The vermicompost derived from Cameroon sheep dung, in comparison to cattle manure-based vermicomposts, contained more N, P and Mg, and less Ca [72], more N, P and K [73], more C, N and P, and less P and Mg [74].

*3.5. Changes in C:N Ratio*

Total carbon content did not differ in both vermicomposts produced (CSD: CSDH) ($p > 0.05$), but it was significantly lower compared to the initial material ($p < 0.05$). A decrease in the organic carbon content is associated with the process of mineralization, which is the transformation of organic carbon into $CO_2$ through the respiration of organisms [19,25].

As a result of the vermicomposting process, a statistically significant increase in nitrogen content in the obtained fertilizers was observed (CSD: from 14.0 to 17.0 g kg$^{-1}$ d.m.; CSDH: from 14.5 to 18.0 g kg$^{-1}$ d.m.). This resulted in a significant reduction in the C:N mass ratio (respectively, from 34.7 and 35.1 to 20.0 in both vermicomposts). Al-Delfi and Abdulkareem [75], studying the composting of poultry manure with or without plant residues, also observed that at the end of the composting process, the total organic carbon decreased, while the total nitrogen increased in contrast with initial values.

The basis of rational nutrient management with the use of composts or vermicomposts in agriculture is to determine C:N ratio and the general forms of mineral ingredients. The C:N ratio in vermicomposts indicates their maturity, so it is commonly used for the assessment of quality of organic fertilizers [20,76–79]. The C:N ratio depends on many factors: e.g., on vermicomposting technology and type of baseline waste. In the vermicomposts produced, C:N ratios ranged from 19 to 22, which indicated the appropriate level of substrate maturity. Coulibaly et al. [18] suggested that vermicompost produced from cattle manure was characterized by a C:N ratio of 27.86; whereas raw manure—of 89.37. In turn, Sharma and Garg [19] processed other types of waste mixtures (composed of cattle

manure, rice straw and paper) and obtained a broader range of C:N ratios (from 12.23 to 38.85), thus demonstrating a varying degree of vermicompost maturity. The results indicated that C:N ratio is more appropriate in vermicompost (12.8) compared to sheep manure (28) [30]. Thus, using vermicompost as a fertilizer would provide better and more facilitative conditions for plants to absorb nitrogen.

### 3.6. Trace Elements

Another important factor determining the usefulness of fertilizers, particularly those that are produced based on organic waste, is heavy metal content [80,81]. The Regulation of the Minister of Agriculture and Rural Development [82] precisely determined the maximum content of nickel, cadmium, chromium, lead, mercury and copper in organic fertilizers. Analyses of the fertilizers produced from Cameroon sheep dung showed a low content of these unwanted chemical elements (Table 2).

**Table 2.** Total content of trace elements in vermicomposts.

| Properties Units | | Features of Vermicomposts | |
|:---:|:---:|:---:|:---:|
| | | CSD | CSDH |
| Cu | | $17.5 \pm 1.9$ [a] | $18.8 \pm 1.8$ [a] |
| Cr | | $5.8 \pm 0.6$ [a] | $5.7 \pm 1.6$ [a] |
| Ni | mg kg$^{-1}$d.m. | <3 * | <3 * |
| Cd | | <0.40 * | <0.40 * |
| Pb | | $14.4 \pm 2.3$ [a] | $13.5 \pm 1.8$ [a] |

CSD and CSDH as in Table 1, * result below the lower range limit of the method [59], aa—no statistically significant differences.

The content of harmful substances, including trace elements (TE), is low over most agricultural areas. Therefore, TE do not have a negative impact on soil functions. However, identification of the areas where there are contaminated soils makes it possible to adopt the environmental risk analysis, remediation measures and mitigation practices. It is an innovative approach, which allows additional ecotoxicological tests and the analysis of the bioavailable contents and the mobile form of pollution to be conducted [83].

The vermicomposting process may affect the reduction in content of trace elements in vermicomposts, which might be the result of bioaccumulation of the elements by earthworms [84,85]. During waste disposal, it comes from the transformation of fractions of trace elements from accessible to inaccessible forms.

During the digestion of organic wastes, the mobile fractions of metals are either accumulated in cutaneous tissues or are bound to low-molecular weight, cysteine-rich metal binding proteins such as metallothioneins (MTs) possessing a high affinity for metals such as Hg, Zn, Ni, Co, Cu and Cd. Most organic acids in vermicompost are also organic chelating agents that absorb micronutrients such as Fe, Zn, Cu and Mn [86].

The increase in assimilable forms of microelements has been noticed—for instance, Bityudskii and Kaidun [87] indicated an increase in the number of assimilable forms of Fe and Mn in earthworm casting, and Devliegher and Verstraete [88] observed that the accessibility of such microelements as Fe, Mn and Cu for plants had increased in the soil in the presence of earthworms (respectively by 6%, 12%, 6%). The best results concerning vermicomposting technology on the solubility of some microelements (Fe, Mn, Cu and Zn) as well as heavy metals (Pb, Cd and Cr) in different combinations of fly ash and organic substances revealed that the introduction of the earthworm *Eisenia fetida*, to different combinations of fly ash and cow dung, resulted in the transformation of a significant number of microelements into bioavailable forms, but the solubility of heavy metals was generally reduced, probably through forming organometallic complexes [89].

Sizmur and Richardson [90] performed an extensive analysis of the bioavailability of potentially toxic elements in vermicomposts and noticed that endogeic and epigeic earthworms increase the mobility of them in the bulk soil, and earthworms from all ecological groups mobilize potentially toxic elements during the passage of soil through the earthworm gut. They also observed an increase in the concentration and uptake of potentially toxic elements by plants growing on soils inhabited by epigeic (mostly *Eisenia fetida*) earthworms. On the other hand, it can alleviate micronutrient deficiencies in degraded soils.

### 3.7. Characteristics of the Earthworm Population

Earthworms are the key organisms that break down organic matter and process nutrients. Dominguez and Edwards [91] described vermicomposting as the process of decomposition composed of interactions between earthworms and microorganisms. These organisms are responsible for the biochemical degradation of organic products, and earthworms are key factors in this process, causing fragmentation and conditioning of the substrate, thus increasing the biological activity of waste [77]. The populations of *Dendrobaena veneta* vermicomposting Cameroon sheep dung (CSD and CSDH) developed well, as shown in Figures 1 and 2.

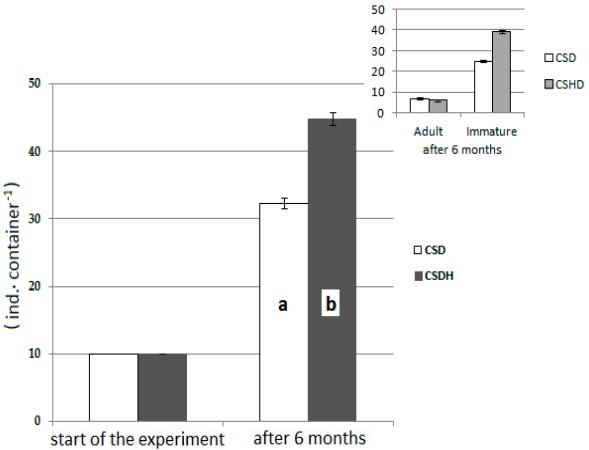

**Figure 1.** The mean number of *D. veneta* earthworms depending on the applied waste ($p < 0.05$). ab—statistically significant differences.

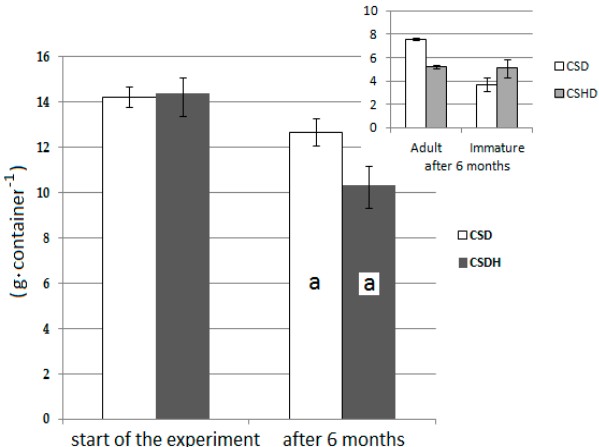

**Figure 2.** The mean biomass of *D. veneta* earthworms depending on the applied waste ($p > 0.05$). aa—no statistically significant differences.

In the experiment, for accelerating the adaptation of sheep dung to the nutritional requirements of earthworms, it had been heated to 105 degrees C, but at the production scale it was achieved by aging dung before vermicomposting [71].

The addition of hay to Cameroon sheep dung (CSDH), as an alternative version to CSD, had a positive impact on the earthworm population of *D. veneta* ($p < 0.05$). Although the two populations did not differ in biomass ($p > 0.05$), the addition of hay significantly enhanced the number of earthworms ($p < 0.05$). This will ensure an obvious profit to the vermicomposter, which is through the possibility to process more material through a greater population of *D. veneta* and/or by the potential sale of these excess earthworms.

## 4. Conclusions and Future Directions

1. Cameroon sheep dung, in a pure form and mixed 1:1 with marginal value hay, can be vermicomposted into a material that has a physico-chemical structure and quality, making it suitable for use as an agricultural fertilizer. As a result of the vermicomposting of organic compounds in them, the "thickening" of mineral components and narrowing of the C:N ratio (carbon loss) took place.

2. The scaling-up of this vermicomposting process to utilize the dung from Cameroon sheep and other ovine stock (in Poland and further afield) could help to consolidate a more circular economy. Further research in this area may prove valuable by linking with small and medium sized farms seeking diversification (post 2020 support from the Rural Development Programme in Poland to be investigated).

3. One major consideration when moving to a farm scale operation is the economics of the process. Before "scaling-up", a comprehensive economic evaluation is required. This should naturally include the removal of dung, production of a more premium quality fertilizer and production of earthworms (potential protein).

**Author Contributions:** Conceptualization, M.G.; data curation, J.K., G.P. and A.M.-P.; investigation, M.G., J.K., G.P., E.H., A.M.-P., K.R.B.; methodology, M.G.; writing—original draft, M.G.; writing—review and editing, M.G., E.H., J.K. and K.R.B. All authors have read and agreed to the published version of the manuscript.

**Funding:** The project was financed under the program of the Minister of Science and Higher Education under the name "Regional Initiative of Excellence" in the years 2019–2022: project number 026/RID/2018/19, funding amount PLN 9 542 500.00.

**Conflicts of Interest:** The authors declare no conflicts of interest.

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
