# Peer review of "Properties of Vermicomposts Derived from Cameroon Sheep Dung"

_applsci, doi:10.3390/app10155048_

Round 1
Reviewer 1 Report
This manuscript has evaluated the feasibility to vermicompost Cameroon sheep dung, in a pure form and mixed with marginal value hay, and determined the chemical properties of the vermicomposts obtained. The effects of these substrates on population dynamics of Dendrobaena veneta were also studied. The experimental design is suitable for testing this topic. I liked such kind of clear, concise and well written manuscript. I believe it will appeal to the readers of Applied Sciences.
Author Response
Response to Reviewer 1 Comments on Ref.: Ms. No. Applied Sciences – 865937
Properties of Vermicomposts Derived from Cameroon Sheep Dung
Please see our response to the points raised by the reviever 1 below.
This manuscript has evaluated the feasibility to vermicompost Cameroon sheep dung, in a pure form and mixed with marginal value hay, and determined the chemical properties of the vermicomposts obtained. The effects of these substrates on population dynamics of Dendrobaena veneta were also studied. The experimental design is suitable for testing this topic. I liked such kind of clear, concise and well written manuscript. I believe it will appeal to the readers of Applied Sciences.
Thank you for these comments.

Reviewer 2 Report
General comments:
The study by Garczyńska et al investigated changes in physicochemical properties of sheep dung before and after vermicomosting and its affect on Dendrobaena veneta populations. I found the introduction to be very well researched and a pleasure to read overall without typos.
The methods are missing some key quality control by not providing information on the replicability and accuracy of the elemental measurements. The Results and Discussion needs to be fixed as the results need to help inform the discussion points. For example, when talking about the C:N ratio, please use the values measured in this study to examine if it will act as a N sink or N source. These simple additions for pH, salinity, macroelements and micronutrients would greatly improve the study.
There are some errors like the error bars on the figures 1 and 2 and missed discussion points to help people using the information from this study to translate it to operational scale. Since the experiment used sheep dung that had been heated to 105 degrees C but this is unlikely to be feasible at the production scale. This should be addressed in the discussion.
Specific comments:
Line 19 – Briefly provide macronutrient concentrations.
Line 20 – p values are typically not provided in abstracts.
Line 22 – Briefly state the trace metal concentrations.
There is plenty of word count to allow for these additional numbers to be provided.
Line 40 – what type of mineral fertilizer as liming and ammonium phosphates are very different in terms of negative environmental impact and costs
Line 49 – Briefly state nutrient concentrations in manures. Particularly as their weight can severely impact their transport and application.
Line 53: Is there any estimate on totally population or abundance of the hair sheep?
Line 77: Provide a citation for composted manure attributes compared to uncomposted manure.
Line 88: There might be an erratic ‘i’.
Line 130 – 133: Please provide quality control and quality assurance for element measurements. In other words, what were the blanks, standard reference material recovery rates, and coefficient of variation.
Line 136 – 138: Why was P and Mg measured with two different methods?
Line 139: Please use atomic symbols consistently throughout the manuscript. Cr, Cu, Pb, and Ni should be abbreviated.
Table 1 should be reformatted so that the table is smaller and units are above their values. Was C/N mol/mol or mass/mass?
Line 148-149: Please rephrase as the sentence is unclear.
Line 157 – 170: This is fine but you need to incorporate results for this study into this discussion on soil humus formation.
Section 3:1 should be combined with the C:N ratio section. No point discussing C without N. Also, what about the information prior to section 3.1, should that be part of a different section?
Lines 191-196: Why did salinity decrease? Did the earthworms enhance their solubility or did they take them up or did they leach from the soils? What ramification does salinity movement have for production scale (think about local rivers if the vermicomposting occurs on bare soil).
Line 203: Please abbreviate Ca consistently throughout the manuscript, especially after first defining it in the introduction.
Line 219: Change “affected” to “caused”.
Line 224: This is confusing, how did salinity decrease but Ca did not change and K increased?
Line 235 – 244: This is fine but you need to better incorporate the C:N results from this study into this discussion on decomposability of the vermicomposted manure. In other words, will this add or deplete N with respect to C?
Lines 275 – 281: This aspect of the study, bioaccessibility of trace metals for plants due to earthworms, could be enhanced by reading information from the study Sizmur and Richardson (2020) Soil Biology and Biochemistry, 107865.
Line 291: The error bars In figures 1 and 2 do not appear to be correct. Please fix.
Author Response
Response to Reviewer 2 Comments on Ref.: Ms. No. Applied Sciences – 865937
Properties of Vermicomposts Derived from Cameroon Sheep Dung
Please see our responses to the points raised by the reviewer below.
General comments:
The study by Garczyńska et al investigated changes in physicochemical properties of sheep dung before and after vermicomosting and its affect on Dendrobaena veneta populations. I found the introduction to be very well researched and a pleasure to read overall without typos.
Thank you
The methods are missing some key quality control by not providing information on the replicability and accuracy of the elemental measurements. The Results and Discussion needs to be fixed as the results need to help inform the discussion points. For example, when talking about the C:N ratio, please use the values measured in this study to examine if it will act as a N sink or N source. These simple additions for pH, salinity, macroelements and micronutrients would greatly improve the study.
Added information
There are some errors like the error bars on the figures 1 and 2 and missed discussion points to help people using the information from this study to translate it to operational scale. Since the experiment used sheep dung that had been heated to 105 degrees C but this is unlikely to be feasible at the production scale. This should be addressed in the discussion.
The experiment was conducted in accordance with the research work procedures. Therefore, the authors of this study applied these procedures and therefore presented the methodology for dealing with manure (drying at 105 ° C). On the other hand, these research procedures do not exclude the extrapolation of the presented studies from the laboratory scale to the production scale, e.g. in farms, where the manure drying process can be performed using other available methods. In the experiment, for accelerating the adaptation of sheep dung to the nutritional requirements of earthworms it had been heated to 105 degrees C but at the production scale it is achieved by aging dung before vermicomposting
Perhaps a line on this could be added.
Specific comments:
Line 19 – Briefly provide macronutrient concentrations.
This has been changed.
Line 20 – p values are typically not provided in abstracts.
This has been removed.
Line 22 – Briefly state the trace metal concentrations.
There is plenty of word count to allow for these additional numbers to be provided.
This has been done.
Line 40 – what type of mineral fertilizer as liming and ammonium phosphates are very different in terms of negative environmental impact and costs
This has been done.
Line 49 – Briefly state nutrient concentrations in manures. Particularly as their weight can severely impact their transport and application.
This has been done.
Line 53: Is there any estimate on totally population or abundance of the hair sheep?
The sheep population is about 100 million, which is about 10% of the world's sheep population. In tropical and subtropical regions, Cameroon sheep make up 60% to 95% of the sheep population. Because they are economical to maintain (they do not require expensive nutrition, expensive procedures such as clipping or hoof correction, and are not susceptible to internal and external parasites), the fastest growth of their breeding is observed in African countries, China and India. They are also becoming popular in Europe.
Line 77: Provide a citation for composted manure attributes compared to uncomposted manure.
Done
Line 88: There might be an erratic ‘i’.
This has been amended.
Line 130 – 133: Please provide quality control and quality assurance for element measurements. In other words, what were the blanks, standard reference material recovery rates, and coefficient of variation.
Presented
The uncertainty for total analytical procedure (including sample preparation) was below the level of 15%. The accuracy was checked using reference materials SRM 2709a ( San Joaquin Soil ) and the recovery (87–102%) was acceptable for determined heavy metals
Line 136 – 138: Why was P and Mg measured with two different methods?
It resulted from the specificity of the determined elements and the technical capabilities of the laboratory, but also in this way in Poland these elements are most often analyzed.
Line 139: Please use atomic symbols consistently throughout the manuscript. Cr, Cu, Pb, and Ni should be abbreviated.
This has been done throughout.
Table 1 should be reformatted so that the table is smaller and units are above their values.
Done
Was C/N mol/mol or mass/mass?
Explained in the text:
C / N was calculated in mass / mass ratios.
Line 148-149: Please rephrase as the sentence is unclear.
Rewritten as „Vermicomposts produced from CSD and CSDH were rich in nutrients (Table 1), but none of these analyzed characteristics were statistically different (p>0.05).”
Line 157 – 170: This is fine but you need to incorporate results for this study into this discussion on soil humus formation.
Done
Section 3:1 should be combined with the C:N ratio section. No point discussing C without N. Also, what about the information prior to section 3.1, should that be part of a different section?
Section 3.1 (Carbon) has been incorporated into the C:N ration section. The info ahead of 3.1 has also been put into its own section.
Lines 191-196: Why did salinity decrease? Did the earthworms enhance their solubility or did they take them up or did they leach from the soils? What ramification does salinity movement have for production scale (think about local rivers if the vermicomposting occurs on bare soil).
This has not been investigated, but possible causes are given in the sentence in the line 194-196: The reason for this finding might be attributed to the easier leaching in vermicompost and to a lesser degree, by the consumption and accumulation of ions in the earthworm biomass.
Line 203: Please abbreviate Ca consistently throughout the manuscript, especially after first defining it in the introduction.
This has been done.
Line 219: Change “affected” to “caused”.
This has been changed.
Line 224: This is confusing, how did salinity decrease but Ca did not change and K increased?
This is also interesting, because in the vermicomposting process, despite the increase in the K content and more or less constant Ca level, the salinity decreased. Salinity was determined on the basis of the specific conductivity of vermicompost suspension in water, so the value of this parameter (and its estimated salinity) is influenced by various ions, including those not tested and not specified in the work. Maybe the primary substrate used for composting was other than K + and Ca 2+ charge carriers - acids or salts, for example in the form of organic compounds - that have been mineralized. This is an interesting starting point for further studies.
Line 235 – 244: This is fine but you need to better incorporate the C:N results from this study into this discussion on decomposability of the vermicomposted manure. In other words, will this add or deplete N with respect to C? supplemented in conclusions
As a result of vermicomposting of organic compounds, "thickening" of mineral components and narrowing of the C: N ratio (carbon loss) was taking place.
Lines 275 – 281: This aspect of the study, bioaccessibility of trace metals for plants due to earthworms, could be enhanced by reading information from the study Sizmur and Richardson (2020) Soil Biology and Biochemistry, 107865.
Done
Line 291: The error bars In figures 1 and 2 do not appear to be correct. Please fix.
Done

Reviewer 3 Report
Line 4: space before „Anna”
Between Lines 6 and 11: the institutions of the authors should be in the same order consequently for all authors, now it is mixed. Suggestion: starting from Department – Institute – University
Line 10: zip code, city, street are missing from author 2
Line 15: „A study…” not „An study…”
Line 18: „… vermicompost obtained xxx characterized…” – something is missing between obtained characterized (which can be…)
Line 27: There are two ones in front of the word „Introduction”
Line 88: „Plectosporium i Verticillium” --- i ? and ?
Line 95: „Determination of soil content for …” – May be it is better: „Determination of the main macroelements (nitrogen, carbon, potassium, phosphorus, calcium, magnesium) of the soil is one of …”
Line 111: I would put the year after the species name: Dendrobaena veneta (Rosa, 1886)
Lines 118, 127-128, 135: „mg·dm-3” -- if you use the „·” symbol between mg and dm-3, then be consequent if you use space around it or not
Line 146: „t-test” not „t test”
Line 148: Table 1. - capital T
Line 150: „Initial biomass” is a little bit confusing for me, because by biomass, I always expect a weight of some material, however, here it is only designating the „initial material” of CSD. I would rather call them „Initial material” and „Final material”.
Line 152: „ … and total of macroelement…” -- macroelement is not plural. Vermincompost – Vermincompost is not plural.
Table 1. could be shrunk a little bit with line spacing 1, and it would fit on one page
Line 151: there was significant difference found in case of K as well, however, it was not listed
Line 171: Section 3.1. should start one Tab to the right
Line 173: space between biomass and (p<0.05)
Line 193: “… have recorder” --- „have recorded”
Line 254: Table 2. - capital T
Line 262: Instead of this “CSD and CSDH as in table 1”, I would write the abbreviations out again, so Table 2, is understandable by itself.
In Table 2, why is Cu in bold?
Line 291: I would write fig 1 and fig 2 with capital F
In Figure 1 and 2, may be the units should be written on the y axis, instead of the figure title. Also, in Figure 1, (ind. container-1) dot is not used, however, in Figure 2 it is used (g ·container-1). It should be consequent.
Regarding Figure 1 and 2, I am surprised that, however, the number of worms increased significantly after 6 month in CSDH, the biomass was lower compared to the starting time? May be, the majority of worms were still juveniles? These species have incubation time (21 to 31 days) and maturation time (82-99 days) according to Zicsi, 1985, thus some individuals could reach maturity within 6 months. It would have been interesting to know the ratio of adult and juvenile after 6 months of the experiment. More discussion on this part would increase the value of the article.
Zicsi, A. 1985. Welche Lumbriciden-Arten eignen sich noch in Europa zum Anlagen von Wurmkulturen zweks Kompostirungsversuche? – Opuscula Zoologica Budapest. 21:137-139.
Line 289: there is an extra space after “earthworm”
Line 383: there is an extra space before “Jahanbakhshi”
References:
Information for Authors:
Journal Articles:
1. Author 1, A.B.; Author 2, C.D. Title of the article. Abbreviated Journal Name Year, Volume, page range.
The Volume of the journal should be in Italics. Please check these.
Author Response
Response to Reviewer 3 Comments on Ref.: Ms. No. Applied Sciences – 865937
Properties of Vermicomposts Derived from Cameroon Sheep Dung
Please see our responses to the points raised by the reviewer below.
Line 4: space before „Anna”
Done
Between Lines 6 and 11: the institutions of the authors should be in the same order consequently for all authors, now it is mixed. Suggestion: starting from Department – Institute – University
Done
Line 10: zip code, city, street are missing from author 2
Done
Line 15: „A study…” not „An study…”
Done
Line 18: „… vermicompost obtained xxx characterized…” – something is missing between obtained characterized (which can be…)
Now corrected. Thank you.
Line 27: There are two ones in front of the word „Introduction”
Done
Line 88: „Plectosporium i Verticillium” --- i ? and ?
Corrected (as noted for Reviewer 2).
Line 95: „Determination of soil content for …” – May be it is better: „Determination of the main macroelements (nitrogen, carbon, potassium, phosphorus, calcium, magnesium) of the soil is one of …”
Changed
Line 111: I would put the year after the species name: Dendrobaena veneta (Rosa, 1886)
This has been added
Lines 118, 127-128, 135: „mg·dm-3” -- if you use the „·” symbol between mg and dm-3, then be consequent (consistent) if you use space around it or not
Thank you. Changed
Line 146: „t-test” not „t test”
Changed
Line 148: Table 1. - capital T
This and other examples have been changed
Line 150: „Initial biomass” is a little bit confusing for me, because by biomass, I always expect a weight of some material, however, here it is only designating the „initial material” of CSD. I would rather call them „Initial material” and „Final material”.
Thank you. Changed
Line 152: „ … and total of macroelement…” -- macroelement is not plural. Vermincompost – Vermincompost is not plural.
Changed
Table 1. could be shrunk a little bit with line spacing 1, and it would fit on one page
Done
Line 151: there was significant difference found in case of K as well, however, it was not listed
Changed
Line 171: Section 3.1. should start one Tab to the right
Changed
Line 173: space between biomass and (p<0.05)
Changed
Line 193: “… have recorder” --- „have recorded”
Changed
Line 254: Table 2. - capital T
As above
Line 262:
In Table 2, why is Cu in bold?
Done
Line 291: I would write fig 1 and fig 2 with capital F
Changed
In Figure 1 and 2, may be the units should be written on the y axis, instead of the figure title. Also, in Figure 1, (ind. container-1) dot is not used, however, in Figure 2 it is used (g ·container-1). It should be consequent.
Changed
Regarding Figure 1 and 2, I am surprised that, however, the number of worms increased significantly after 6 month in CSDH, the biomass was lower compared to the starting time? May be, the majority of worms were still juveniles? These species have incubation time (21 to 31 days) and maturation time (82-99 days) according to Zicsi, 1985, thus some individuals could reach maturity within 6 months. It would have been interesting to know the ratio of adult and juvenile after 6 months of the experiment. More discussion on this part would increase the value of the article.
Zicsi, A. 1985. Welche Lumbriciden-Arten eignen sich noch in Europa zum Anlagen von Wurmkulturen zweks Kompostirungsversuche? – Opuscula Zoologica Budapest. 21:137-139.
Changed
Line 289: there is an extra space after “earthworm”
Removed
Line 383: there is an extra space before “Jahanbakhshi”
Removed
References:
Information for Authors:
Journal Articles:
1. Author 1, A.B.; Author 2, C.D. Title of the article. Abbreviated Journal Name Year, Volume, page range.
The Volume of the journal should be in Italics. Please check these.
Thank you. Changed

Round 2
Reviewer 2 Report
The authors have done a good job addressing my comments and I believe the manuscript should be accepted in its revised form. I thank the authors for addressing my comments well and hope that the research can be implemented.